# A Review of High-Temperature Resistant Silica Aerogels: Structural Evolution and Thermal Stability Optimization

**DOI:** 10.3390/gels11050357

**Published:** 2025-05-13

**Authors:** Zhenyu Zhu, Wanlin Zhang, Hongyan Huang, Wenjing Li, Hao Ling, Hao Zhang

**Affiliations:** Aerospace Research Institute of Special Material and Processing Technology, Beijing 100074, China

**Keywords:** silica aerogel, thermal stability, structural evolution, high-temperature aerogel

## Abstract

Silica aerogels exhibit exceptionally low thermal conductivity and a low apparent density, as they are unique porous nanomaterials. They are extensively used in thermal insulation in terms of aerospace and building construction, adsorption processes for environmental applications, concentrating solar power systems, and so on. However, the degradation of the silica aerogel’s nanoporous structure at high temperatures seriously restricts their practical applications. Through a comprehensive review of the high-temperature structural evolution and sintering mechanisms of silica aerogels, this paper introduces two strategies to enhance their thermal stability, including heteroatom doping and surface heterogeneous structure construction. In particular, atomic layer deposition (ALD) of ultra-thin coatings on silica aerogel holds significant potential for enhancing thermal stability, while preserving its ultra-low thermal conductivity.

## 1. Introduction

Aerogels have unique material properties, characterized by their highly porous microstructure, possessing a network of loosely packed and bonded particles or fibers [1,2,3]. They are useful for a broad range of applications, in fields such as aerospace, energy, environmental protection, transportation, buildings, and medical facilities. They have garnered significant attention in recent decades [4,5,6].

Silica aerogels are widely investigated and used, due to their extremely high porosity (over 90%), low density (0.03–0.40 g cm^−1^), high specific surface area (500–1200 m^2^/g), and low thermal conductivity (0.012–0.024 W m^−1^K^−1^). Silica aerogels also exhibit a low refractive index (1.00–1.08), sound velocity (100 m s^−1^), and dielectric constant (1.0–2.0) [7]. Owing to their unique properties, silica aerogels demonstrate significant potential in a diverse range of applications that require thermal insulation, fire prevention, light transmittance, noise reduction, and adsorption properties [8]. In addition, superhydrophobic silica aerogels exhibit remarkable efficacy as reusable absorbents for oils and organic liquids, boasting a high uptake capacity and efficiency [9]. The utilization of silica aerogels as photoanodes for dye-sensitized solar cells is attributed to their exceptional high surface area and porosity [10]. The low dielectric constant of silica aerogels makes them suitable for use as intermetal dielectric shielding materials in microelectronic devices [11].

Silica aerogels are presently the predominant material used in industrial thermal insulation, among the different classes of aerogels [12]. The aerogel structure is composed of a cross-linked silica-particle backbone that provides support for a highly mesoporous (2–50 nm) 3D continuous network, consisting of aggregated particles. The solid and gaseous thermal conductivity of aerogels is significantly reduced due to their low solid volume fraction and pore sizes that are smaller than the mean free path in air, resulting in thermal conductivity lower than that of air at room temperature [7]. And radiative thermal conductivity is also significantly reduced, due to the high absorptivity of silica particles in infrared (IR) wavelengths [13]. The low thermal conductivity of silica aerogels makes them suitable for insulation purposes in building construction [14,15], concentrating solar power devices [16], and aeronautics and aerospace systems [17].

However, the limited thermal stability of silica aerogel significantly hinders its application in high-temperature scenarios (Figure 1), such as aerospace thermal insulation, concentrating solar power systems, and solid rare isotope catchers, despite the advantages of its superior thermal insulation properties [16]. Pure silica aerogel exhibits long-term stability up to 650 °C. As the temperature continues to rise, the densification and particle aggregation of the aerogel become predominant, which can result in detrimental effects to the material’s microstructure and thermal conductivity. At a higher temperature than the glass transition temperature of the aerogel, extreme densification and particle aggregation induce significant alterations in the nanostructure within a few hours. Consequently, the nanoporous structure of SiO_2_ aerogels collapses and disintegrates, leading to a failure in terms of their thermal insulation performance [16,18,19].

The deployment of traditional silica aerogels has been constrained by concerns regarding their thermal stability. Improving the thermal stability of silica aerogels is critical for broadening their scope of application, particularly in industries that require thermal insulation in extreme conditions. Various strategies have been explored to overcome this limitation, including chemical doping, surface modification, and the use of composite materials. Wang et al. [20] comprehensively reviewed the five types of high-temperature aerogels, including polyimide-based, zirconia-based, silica-based, alumina-based, and carbon-based aerogels. In the section on silica-based aerogels, the authors provide a concise review of performance optimization, referring to thermal conductivity, strength, toughness, and thermal stability. The focus is primarily on composite aerogels. This review mainly discusses the improvements in the thermal stability of the aerogel structure itself, including doping and surface modification. In particular, the structural evolution and thermal insulation failure mechanism of silica aerogels at high temperatures are introduced. Based on this mechanism, the methods developed to improve the thermal stability of silica aerogels were systematically examined, with a focus on the underlying mechanisms, recent advancements, and challenges.

## 2. Structural Evolution and Sintering Mechanism at High Temperatures

Silica aerogels exhibit exceptionally low thermal conductivity, high porosity, and a low apparent density, which is why they are used for super-thermal insulation. However, pure SiO_2_ aerogel composites become sintered at high temperatures (e.g., over 800 °C). The backbone of silica aerogel is composed of primary particles, which are silica nanoparticles. Furthermore, these primary particles aggregate to form secondary particles, as illustrated in Figure 2a. The pearl necklace-like backbone matrix of silica aerogels, with a highly porous microstructure, is the result of the aggregation of secondary particles [7,8]. The sintering process involving silica aerogels, driven by a reduction in the surface energy, intensifies particle aggregation and skeleton coarsening, leading to the densification and degradation of their nanopore structure. This structural degradation results in an increase in solid thermal conductivity, thereby compromising the thermal insulation performance of silica aerogels [21]. Therefore, investigating the structural evolution of silica aerogel at high temperatures, elucidating the underlying change rule, and uncovering the sintering mechanism, has significant guiding implications for enhancing the thermal stability of silica-based aerogels.

### 2.1. High-Temperature Structural Evolution

Back in 1996, Kuchta et al. described the changes occurring during the heating of a SiO_2_ aerogel, synthetized from tetramethoxysilane (TMOS), in the temperature range from 20 to 1000 °C [22]. The results indicate that there is no dimensional change in the aerogel in the temperature range from room temperature to 200 °C. From 200 °C to 500 °C, the aerogel sample exhibits a gradual and progressive shrinkage (~1% of its original length). Above 600 °C, the shrinkage of the SiO_2_ aerogel becomes more pronounced and can reach values of 30–40%, which depends on factors such as sample handling prior to measurement or the heating rate during the measurement. The density of the sample increases proportionally with the changes in its dimensions and volume, while the specific surface area decreases.

Many researchers have studied the structural evolution of silica aerogels. The investigation of the evolution of the nanostructure in silica aerogels at elevated temperatures typically involves conducting an isothermal heat treatment. In detail, the aerogel is exposed to a specific temperature and, subsequently, analyzed for any resulting changes in its structural and morphological characteristics upon returning to room temperature.

Emmerling et al. [21] used the small-angle X-ray scattering technique to monitor the changes in the characteristics of the nanostructure of aerogel networks at 1050 °C. During the initial stage, the preferential densification of large pores in the silica aerogel was observed. Bouaziz et al. [23] investigated the phase transition process of silica aerogels after heat treatment, by using scanning electron microscopy (SEM). The results demonstrated that the phase transition process of aerogels after heat treatment was homogeneous. Wagh et al. [24] studied the pore size distribution in silica aerogels after the heat treatment process, via SEM. The investigation revealed that the pore radius and pore size distribution exhibited an increase up to a temperature of 800 °C. However, beyond this temperature, there was a narrowing in the pore size distribution accompanied by a decrease in the pore radius. Buscarino et al. [25] used atomic force microscopy (AFM) to investigate the change in particle size of amorphous SiO_2_ nanoparticles during the sintering process (Figure 2c). Using Raman and infrared spectroscopy, it was found that the Si-O-Si bond angle and Si-O ring size of silica nanoparticles are smaller than that of bulk silica particles and this difference becomes smaller after heat treatment. These results indicated that thermal relaxation occurred involving the silica nanoparticles during the sintering process. Strobach et al. [16] investigated the high-temperature stability of transparent silica aerogels. Silica aerogel samples were annealed at temperatures of 400 °C, 600 °C, and 800 °C, for several months, to study the relative changes in the nanostructure, effective thermal conductivity, and solar transparency (Figure 2b). The results revealed that at both 400 °C and 600 °C, the temperature-dependent changes reached a plateau within a month. However, at an annealing temperature of 800 °C, the samples exhibited structural instability and their properties quickly degraded.

**Figure 2 gels-11-00357-f002:**
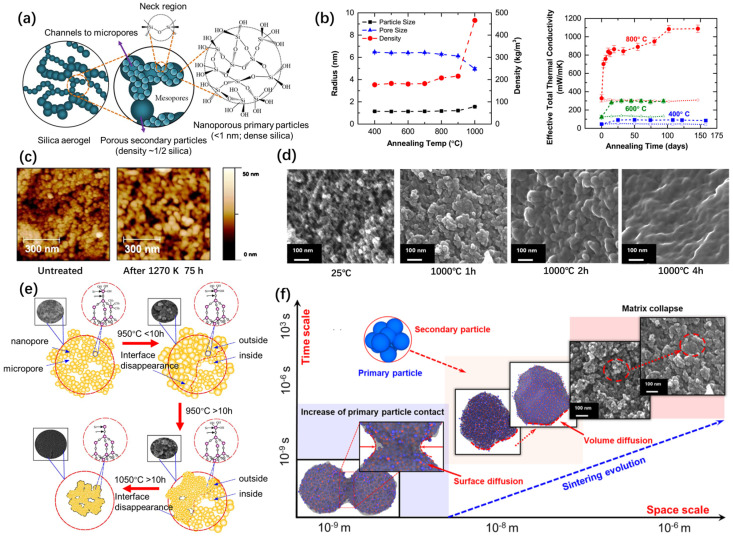
High-temperature structural evolution of silica aerogels. (**a**) Microstructure of pure silica aerogels. Adapted with permission from [12], copyright 2020, Elsevier. (**b**) Average pore size, particle size, and overall density change in silica aerogels after 1 h annealing and effective thermal conductivity of silica aerogels as a function of the annealing time [16]. (**c**) AFM images obtained of the nanoparticles in silica aerogel before and after heat treatment. Adapted with permission from [25], copyright 2011, Elsevier. (**d**) In situ TEM images of nanostructure evolution in silica aerogels. Adapted with permission from [26], copyright 2020, Elsevier. (**e**) Schematic of the network structure evolution and corresponding SEM images, along with the heating time for three steps. Adapted with permission from [27], copyright 2017, Elsevier. (**f**) High-temperature effects on silica aerogel and sintering evolution of silica aerogel, with space and time scales. Adapted with permission from [28], copyright 2022, Elsevier.

The small-angle X-ray scattering (SAXS) technique can characterize the structure of aerogels at “small” length scales (0.1–10 nm) [29]. At the nano-level scale, aerogels can be described as self-similar (fractal) materials. Fractal parameters are used to characterize the structure of aerogels, such as the particle size, *a* (typically in the range of 1 nm); the mean size of the fractal clusters, *ξ* (coherence length, about several tens of nanometers); and the fractal dimension, *D*.

Marlière et al. [29] used two complementary methods, ultra-small-angle X-ray scattering (USAXS) in the reciprocal space and AFM in the real space, on silica aerogels, to investigate the densification process of silica aerogels. The findings revealed a sub-micrometric superstructure at a large scale (~100 nm) for the first time, consisting of aggregates of clusters. Notably, these clusters exhibited fractal characteristics during the initial stages of sintering.

The evolution of the superstructure through the sintering process was further investigated. Silica particles undergo an increase in size during this process, leading to cluster contraction and, ultimately, resulting in the collapse of aggregate structures.

Huang et al. [27] investigated the structure of silica aerogels subject to high temperatures, from 950 °C to 1200 °C, based on the combination of various characterization techniques, such as SEM, Fourier transform infrared (FTIR) spectroscopy, X-ray diffraction (XRD) patterns, and Brunauer–Emmett–Teller (BET) analysis. It was obtained that the structural changes in silica aerogels can be divided into the following three steps, as shown in Figure 2e: the expansion of primary particles at the surface (step I), pore collapse and atrophy of the primary particles at the sample surface (step II), and pore collapse and atrophy of the primary particles inside the backbone of the silica aerogel (step III). During step III, the pore structure of the silica aerogels was completely destructed, while the density exhibited a rapid increase.

Although numerous investigations have previously examined the structural evolution of silica aerogels during heat treatment, a more comprehensive understanding of their sintering behavior can be achieved through real-time observations of their structure and morphology. These observations offer a robust experimental and theoretical foundation for enhancing the thermal stability of aerogels. Cai et al. investigated the evolution of the nanostructure of silica aerogels [26] (Figure 2d) and composites reinforced with mullite fibers [30] via in situ TEM, with a rapid heating treatment, in the range from 600 °C to 1300 °C. The shrinkage of aerogels and composites primarily occurs during the initial stages of the heating process. This shrinkage is predominantly attributed to the agglomeration of small particles within the aerogels at elevated temperatures, with no discernible further reduction in size as the duration of the heating increases.

According to the findings from the aforementioned studies, a comprehensive summary can be provided regarding the structural modifications of silica aerogels during heat treatment at various temperatures. The residual organic groups on the surface of silica aerogels undergo oxidation primarily below 600 °C, leading to a substantial weight loss in the aerogel samples. However, this oxidation process does not significantly alter the nanostructure of silica aerogels. He et al. [31] conducted TG (thermogravimetric)–IR analysis to study the effects of high temperatures on hydrophobic silica aerogels. C–O, −OH, and CO were detected during the pyrolysis of silica aerogels in air. The pyrolysis process can be divided into three steps, which are the hydroxylation of –CH_3_, the splitting of the alcoholic hydroxyl, and CO oxidization. The shrinkage of the aerogel is not due to the oxidative decomposition of unreacted organic groups [30]. However, the decomposition of the organic fragments increases the specific surface area of the material, which further increases the surface energy of aerogels and facilitates a greater sintering driving force.

A distinct alteration in the structure of a silica aerogel was observed (Figure 2f) when the heat treatment temperature exceeded 600 °C. The evolution of the nanostructure and sintering process of silica aerogels at elevated temperatures can be categorized into three stages. Within the range of 600 °C to 800 °C, the collapse of small pores occurs, while larger pores undergo expansion. The particle size of aerogels will initially experience a slight increase, followed by a subsequent absence of any significant alterations. The backbone matrix remains complete throughout the heating process. However, alterations in the density and porosity of aerogels do occur, leading to an augmentation of their thermal conductivity. With the increase in temperature between 800 °C and 1000 °C, the aggregation of the primary particles is intensified and the secondary particles become larger gradually. The particle size distribution is gradually reduced. The mean pore size will decrease and the size distribution will narrow. And the porous network structure of silica aerogels begins to collapse after a long time of sintering. When the heat treatment temperature exceeds 1000 °C, the fusion between the secondary particles is accelerated and the pores collapse rapidly. The porous skeleton structure of silica aerogels will be damaged, with the pore structure disappearing gradually. At this temperature, the aerogel will continue to shrink and will eventually become dense.

### 2.2. Sintering Driving Forces and Models

The structural deterioration of nanoporous aerogel materials at high temperatures occurs during the solid-phase sintering process. Sintering is a process involving the consolidation of material particles via heat treatment at a temperature that is lower than the absolute melting point. The primary driving force that propels the sintering operation is the reduction in energy on the surface of the material [32,33]. More realistic attributes in terms of the surface of the nanoparticles at the sintering temperature are rather the development of a liquid or liquid-like layer [28]. During the heat-treatment process, the reduction in the surface area as the particles are bonded together is the primarily reason for the coarsening of the skeleton and the increase in thermal conductivity of nanoparticle aerogels at high temperatures.

The surface energy is affected by the reduction in particle size. Thermodynamically, the total Gibbs free energy of a particle is the sum of the Gibbs free energy of the bulk particles and the Gibbs free energy of the particle’s surface. The specific surface area increases rapidly with the decrease in the particle size. Therefore, the contribution of the surface component becomes prominent as the particle size decreases [34]. It is expected that with the increased surface area of the nanoparticles, an increase in the reactivity of the particles will be achieved. When the particle size reaches the nanometer scale, a significant proportion of atoms in the particles are situated at the surface, resulting in nanoparticles possessing exceedingly high surface energy. Accordingly, the total surface area of more refined powder grains tends to increase, thereby significantly enhancing the driving force of the entire sintering process.

Mass transportation occurs in six different ways according to the sintering mechanisms shown in Figure 3a, which are surface diffusion, lattice diffusion from the surface, vapor transport, grain boundary diffusion, lattice diffusion from the grain boundary, and plastic flow via dislocation motion [32]. The first three mechanisms are non-densifying mechanisms, producing microstructural changes, without causing shrinkage. In contrast, the last three mechanisms involve a densification process by removing material from the grain boundary region, resulting in shrinkage. The sintering process typically involves multiple diffusion mechanisms. To minimize the surface energy, atoms near the particle boundaries migrate and facilitate necking growth (Figure 3b). The tensile stress σ of two nanoparticle surfaces that are in contact with each other can be described using the following formula:$$\sigma =-\frac{\gamma }{\rho }$$
where *γ* is the surface tension and *ρ* is the radius of curvature. Notably, the concept of surface free energy is distinct from that of surface tension. Surface energy refers to the minimum work to form a unit area of the surface by a process of division, while surface tension represents the tangential stress (force per unit length) within the surface layer. From the above formula, it can be concluded that the tensile stress acting on the neck points to the outside of the necking. The diffusion of the liquid-like layer in response to the sintering stress results in necking growth and the fusion of aerogel particles at high temperatures.

The variation in the surface curvature results in distinct saturated vapor pressures across different regions of the system, thereby inducing a propensity for mass transfer through the vapor transport mechanism. The smaller the particle size, the larger the vapor pressure ratio. Hence, the gas phase during the mass transfer of nanomaterials at high temperatures cannot be ignored. The vapor pressure ratio at a high temperature and on the particle surface can be calculated using the following formula:$$Ln\left(\frac{P_{1}}{P_{0}}\right)=\frac{\gamma }{dRT}\left(\frac{1}{\rho }+\frac{1}{r}\right)$$
where *P*_1_ is the vapor pressure of the sintered necking, *P*_0_ is the vapor pressure at the surface of the spherical particle, *ρ* is the curvature radius of the sintered necking, *r* is the radius of the contact plane of the necking, and *γ* is the surface tension of the material. Moreover, *d*, *R*, and *T* are the density, gas constant, and absolute temperature, respectively. Accordingly, the mass will evaporate from the convex surface of the particle (positive radius of curvature), resulting in high saturated vapor pressure, will transfer through the gas phase, and will condense in regard to the concave neck (negative radius of curvature), resulting in low saturated vapor pressure, and the neck will be gradually filled.

In order to deeply understand the structural evolution of silica aerogels during the heat treatment process, numerous researchers have proposed diverse sintering models based on numerical experimental results and simulations. The viscous flow theory is the basic sintering model of silica aerogels.

In essence, Frenkel was the first to apply rheological methods to study the sintering process [40,41]. Frenkel’s sintering was derived in two model problems as a consequence of “viscous” flow: the joint sintering of two equal spherical particles and the shrinkage of pores in an infinite viscous medium. During the sintering process of silica aerogels, the viscosity of the aggregated nanoparticles is reduced, enabling relative movement between the particles, thereby achieving overall structural relaxation [16,21,25,42], which results in a shrinkage of both the mean pore size and the overall volume. With a further increase in temperature, the viscosity of the particles becomes lower such that the particle aggregation and densification dominate the structural evolution.

This direction was further developed by [43]. Based on the idea of cylindrical powder particles, the viscous sintering model of amorphous materials was introduced into the viscous flow theory. This model was applied to simple cell geometries to describe the sintering process of open-pore bodies, including silica aerogels (Figure 3c), so-called cell models [36,44,45]. The sintering process proceeds by increasing the radius-over-length ratio with a constant total mass. Moreover, Scherer derived that the kinetic equation of the densification rate was d for a cubic cylindrical array driven by viscous flow, with surface energy reduction. Based on the initial parameter in terms of the pore size distribution in aerogels, the evolution of densification with the change in the density, surface area, and mean pore size were accurately predicted using the theory of viscous sintering.

Subsequently, some scholars amplified and further developed the theory by Scherer. Sempéré et al. [37,46] proposed a simple sintering model to describe connected fractal aggregate materials. The densification at small scales in the model is described by a decrease in the upper cutoff length (Figure 3d), *ξ*, accompanied by an increase in the lower cutoff length, *a*, in order to conserve the total mass of the system. This theory satisfactorily explained silica aerogel densification, using a significant number of experiments.

Rosa-Fox et al. [47] proposed an aggregation model, depicting the structure of aerogels as a hierarchical arrangement of spherical units. The model incorporates both the hierarchical ratio, which is the size of a cluster in terms of the size of the elementary unit, and the clustering index, which is the number of elementary units in a cluster. The network connectivity during the heating process was characterized by SAXS and BET analysis.

Jullien and Olivi-Tran [38,48,49] developed a scaling theory to explain the sintering process of fractal materials. The densification at small scales is described by a decrease in the upper cutoff length, with an increase in the lower cutoff length, ensuring the conservation of the system’s total mass (Figure 3e). The sintering process can be modeled in two steps. The “dressing” step, where all particles are replaced by overlapping larger particles. And the “contraction” step, where adequate length rescaling is performed in order to conserve the total mass (the total area in Figure 3e).

Phalippou et al. [39] proposed a microscopic model for the change in the nanoporous structure of aerogels, as a result of thermal sintering and isostatic compression. SAXS measurements were performed on aerogels exhibiting a fractal geometry in order to follow the structural evolution of aerogels (Figure 3f). The analysis of the pore size distribution provided evidence that pressure induces the collapse of the largest pores, while sintering affects all the pores.

In addition, some scholars have revealed the nanoscale mechanisms of heat and mass transfer in silica aerogels through molecular dynamics methods [28,50,51,52,53].

An in-depth understanding of the evolution of the nanostructure of silica aerogels in regard to heating and sintering mechanism provides a solid theoretical and experimental basis for improving the thermal stability of aerogels. Proceeding from this mechanism, effective strategies for improving the thermal stability of aerogels can be derived. Increasing the melting point of the material can reduce the viscosity of the particles at sintering temperature, thus inhibiting the deterioration of the structure. Reducing the surface energy through heterogeneous interfaces limits mass transfer during high-temperature diffusion.

## 3. Methods for Improving the Thermal Stability of Silica Aerogels

Based on the high-temperature structural evolution behavior and sintering mechanism of silica aerogels, it can be deduced that reducing the content of large pores or increasing the size of the secondary particles could enhance the thermal stability of aerogels. Nevertheless, the thermal conductivity of silica aerogels will inevitably increase, which does not satisfy the fundamental requirements of thermal insulation materials. Currently, the approaches for improving the thermal stability of silica aerogels are to introduce heat-resistant phases through heteroatom doping and to construct heterogeneous structures on the surface of the silica aerogel backbone. Two strategies for improving the thermal stability of silica aerogels are elaborated below (Figure 4).

### 3.1. Heteroatom Doping

At present, the main methods in terms of doped silica aerogel include immersion, chemical vapor deposition, chemical vapor osmosis, and the in situ sol–gel method [54,55,56,57,58]. The in situ sol–gel method is widely employed for the synthesis of doped silica aerogels. The preparation process involves the addition of a precursor or nanoparticle solution involving a doping agent to the silicon dioxide precursor solution and then to add a catalyst to obtain a mixed solution. Alternatively, a mixed solution was obtained by mixing the prepared silicon dioxide solution with the metal precursor solution. Through processes including gelation, surface modification, and drying, the corresponding component-doped aerogel materials can be obtained.

#### 3.1.1. Alumina-Doped Silica Aerogels

The doping of other ceramic thermally resistant phases is a viable approach to improve the thermal stability of silica aerogels. The incorporation of a refractory phase with enhanced dimensional thermal stability (such as alumina) into silica aerogel matrices, as demonstrated in previous studies [59,60,61], leads to a significant enhancement in the thermal stability of silica aerogels. Aluminum-doped silica aerogels maintain low thermal conductivity. In the study by Ling et al., silica-based aerogels doped with 1.28 to 7.46 wt% alumina exhibited thermal conductivities of 0.030 to 0.039 W m^−1^ K^−1^ at room temperature, which increased to 0.057 to 0.074 W m^−1^ K^−1^ at 800 °C [62]. Many researchers have conducted in-depth and sufficient research on Al_2_O_3_–SiO_2_ aerogels in terms of the doping amount, reaction raw materials, and sol–gel drying methods. There are basically two routes to prepare silica–alumina aerogels, according to the type of precursor used, as follows [63]:

The organic aluminum precursor-based route. The precursors employed in this route are primarily organosilicons (TEOS and TMOS) and organic aluminum compounds (aluminum isopropoxide and aluminum tri-sec-butoxide), while the sol–gel method (Figure 5a) combined with the supercritical drying technique was adopted [64,65,66]. During the preparation of Al_2_O_3_–SiO_2_ aerogels, the hydrolysis of organic aluminum compounds occurs significantly faster than that of organosilicons, resulting in a non-uniform solution due to sedimentation that occurs before gelation. It becomes necessary to introduce chelating agents to inhibit the hydrolysis of organic aluminum compounds [65,66,67] or to promote the hydrolysis process of organosilicons [68], thereby achieving a synchronized reaction process between these two precursors. As early as 1993, Komarneni et al. [59] carried out a study on the Al_2_O_3_–SiO_2_ system aerogel, using Al doping amounts of 1% and 10%. Tetramethoxysilane (TMOS) and boehmite were used as the starting materials. The presence of alumina as a refractory phase hindered their densification, resulting in surface areas ranging from 500 to 600 m^2^/g, with mesopore diameters of approximately 6 nm, after being heated at 1000 °C;The inorganic aluminum precursor-based route. In this route, inorganic aluminum salts, mainly including aluminum chloride and aluminum nitrate, are used as precursors, along with organosilicon [69,70,71,72]. In order to reduce the cost, natural or industrial waste containing silicon and aluminum can be adopted as precursors. Rutiser et al. [60] prepared aerogels from silica combined with 1 to 10 mol% of minerals containing aluminum, such as kaolinite, montmorillonite, boehmite, and mullite. When annealed at 1000 °C, alumina-doped silica aerogels still retained a very high surface area compared to sintered aerogels prepared from silica only. Specific surface areas of up to 425 m^2^/g were achieved by the 1% kaolinite aerogels after 8 h of heat treatment.

In addition to supercritical drying, alumina-doped silica aerogel can also be prepared by subcritical drying and ambient pressure drying. Aravind et al. [61] synthesized silica–alumina aerogels using the subcritical method for the first time, with compositions ranging from 5 to 25 wt.% of alumina. TEOS served as the precursor for silica, while boehmite was used for the alumina. The specific surface area of the silica–alumina aerogel with 5 wt.% alumina content at 500 °C was 796 m^2^/g. When the alumina content reaches 25%, the pore structure of silica aerogels can be maintained at 1200 °C. The linear shrinkage of the 25 wt.% alumina-doped sample was only 15%, while the linear shrinkage of silica aerogels is as high as 30% at same temperature.

**Figure 5 gels-11-00357-f005:**
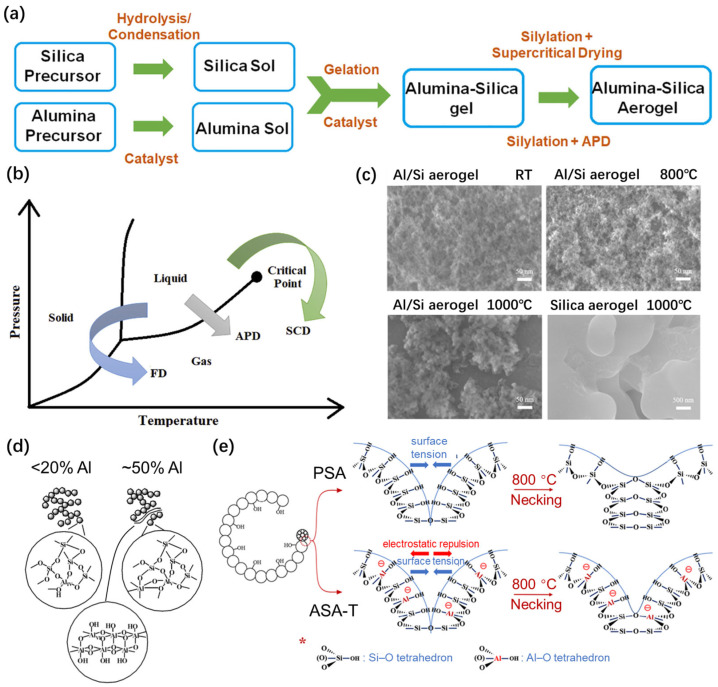
Alumina-doped silica aerogels with modified thermal stability. (**a**) General steps in the synthesis of alumina–silica aerogels using sol–gel technology. (**b**) Freeze drying (FD), ambient pressure drying (APD), and supercritical drying (SCD) techniques applied to aerogels. Adapted with permission from [73], copyright 2020, Elsevier. (**c**) SEM images of alumina–silica aerogels and silica aerogels treated at different temperatures. Adapted with permission from [63], copyright 2022, Elsevier. (**d**) Schematic diagram of alumina-doped silica aerogel structure depending on the aluminum content. Adapted with permission from [64], copyright 2017, Elsevier. (**e**) Schematic diagram of the mechanism of enhanced thermal stability of Al-doped silica aerogels compared to pure silica aerogels. Adapted with permission from [63], copyright 2022, Elsevier.

Yao et al. [63] prepared an alumina-doped silica aerogel using the sol–gel method and ambient pressure drying technique (Figure 5b). The silicon source was cheap industrial water glass and the aluminum source was aluminum chloride. The researchers further measured the thermal stability of the aerogel. The results showed that the material could still maintain a high specific surface area of 179.5 m^2^/g, much higher than that of pure silica aerogel (Figure 5c), after annealing at 1000 °C, although the aluminum content of the aerogel was only 0.18 wt%.

Therefore, the superior thermal stability of silica aerogels, with effective doping of alumina in the lattice, originated from the enhanced thermal stability of Al-O-Si bonds in the Al-contained region [64] (Figure 5d) and the higher melting point of the material. Furthermore, the increase in the aluminum content resulted in the formation of an Al-rich phase, such as that involving plate-like pseudoboehmite particles. In additional, the replacement of Si^4+^ by Al^3+^ within the SiO_2_ lattice induced localized negatively charged centers on the nanoparticle surface (Figure 5e), thereby generating electrostatic repulsion between Al-O tetrahedrons that were dispersed throughout the entire aerogel backbone matrix [63]. The electrostatic repulsion restrained the necking process of silica nanoparticles, driven by their intrinsic surface energy. At the same temperature, the fusion rate of the Al-doped silica skeleton was significantly slower than that of the pure silica aerogel. Consequently, the shrinkage of the sample was reduced and the thermal stability was enhanced.

#### 3.1.2. Zirconia-Doped Silica Aerogels

Zirconia aerogels are potential candidates for high-temperature applications, attributed to the high melting point of ZrO_2_ at 2715 °C. However, pure zirconia aerogels at high temperatures experience a volume change, with phase transformation [74]. The phase transformation process greatly reduces the structural stability of zirconia aerogels at high temperatures, resulting in the collapse of the pore structure. ZrO_2_ is suitable as a temperature-resistant phase doped silica aerogel.

Hu et al. [75] investigated the thermal stability of a ZrO_2_–SiO_2_ aerogel. The particle growth was inhibited by ZrO_2_ upon firing. Meanwhile, SiO_2_ could act, in terms of a heterogeneous phase, to inhibit the phase transition and crystallization of ZrO_2_. Therefore, the pore structure of the ZrO_2_–SiO_2_ aerogel can remain stable at 1000 °C and has a high specific surface area of 203.5 m^2^/g and an appreciable pore volume of 0.721 cm^3^/g after heating.

Wang et al. [76] prepared a ZrO_2_–SiO_2_ aerogel using sol–gel methods, using ZrOCl_2_·8H_2_O as the zirconium source and mercaptosuccinic acid–triethoxyvinylsilane as the gel initiator. The content of SiO_2_ and ZrO_2_ in the aerogel skeleton is controllable. The experimental results from the thermal stability study show that the aerogel could still maintain the amorphous state after annealing at 600 °C in air and the specific surface area remained at 53.4 m^2^/g when the temperature went up to 1000 °C

Liu et al. [77] synthesized a series of ZrO_2_–SiO_2_ aerogels via the sol–gel method and investigated the nanopore collapse, volume shrinkage, and crystallization behavior of the materials after heat treatment. The results show that the uniform crosslinking of ZrO_2_ and SiO_2_ clusters enhances the thermal stability of the material. The mechanism involves the preparation process, namely that the weakly branched ZrO_2_ and SiO_2_ clusters tend to be freely entangled.

Yu et al. [78] synthesized a ZrO_2_–SiO_2_ aerogel with different particle sizes via the solvent-thermal aging (STA) method (Figure 6a). The effects of temperatures ranging from 60 °C to 210 °C on the microstructure and the thermal stability of the ZrO_2_–SiO_2_ aerogel were studied. After heat treatment at 1000 °C for 1 h, the minimum shrinkage rate of the ZrO_2_–SiO_2_ aerogel sample aging at 210 °C was only 9.3%, the specific surface area retained 59.2% of the original, and the pore volume was retained up to 90.8%.

Han et al. [79] synthesized dimethyl-diethoxysilane-modified ZrO_2_–SiO_2_ aerogels (DDS/ZSAs) (Figure 6b). The pore volume increased from 1.65 cm^3^/g to 3.21 cm^3^/g and the thermal conductivity decreased from 0.03013 Wm^−1^K^−1^ to 0.02332 Wm^−1^K^−1^ at room temperature. The DDS/ZSA aerogel exhibits the lowest thermal conductivity (0.035 W m^−1^ K^−1^) when the dilute ammonia content is appropriate [81]. The particle growth caused by hydroxyl condensation was inhibited during firing. The increase in the number of Si-O-Zr bonds inhibited the growth of t-ZrO_2_ and the phase transition temperature of the aerogel was significantly improved. Therefore, it still had a high specific surface area (259.6 m^2^/g), a high pore volume (1.51 cm^3^/g), and low thermal conductivity (0.04206 Wm^−1^K^−1^) after heat treatment at 1000 °C.

#### 3.1.3. Silica Aerogels Doped with Other Elements

Sousa et al. [80] prepared porous Cu–SiO_2_ aerogels using the sol–gel method, using tetraethoxysilane (TEOS) with CuSO_4_ and CuCl, with copper content of 1 and 5 mol%. The structural integrity of all the samples remained stable up to 900 °C, while a significant densification process occurred at 1100 °C. After the annealing process at 900 °C, the surface area of the aerogel doped with CuCl was 40 m^2^/g and the specific surface area of the samples doped with CuSO_4_ was 163 m^2^/g (Figure 6c). After the same treatment, the specific surface area of the pure silica aerogel was only 11 m^2^/g.

Li et al. [82] successfully prepared a high-specific-surface-area copper-doped silica aerogel using the sol–gel method under atmospheric pressure. During the synthesis process, N, N-dimethylformamide (DMF) was mixed with the solution, which consisted of copper nitrate (Cu(NO_3_)_2_ 3H_2_O) and tetraethyl orthosilicate (TEOS). The results show that the copper-doped aerogel exhibited a pore diameter ranging from 2 to 15 nm. The specific surface area of the aerogels with 5% Cu-doping content for 3 h was 171.9 m^2^/g after heat treatment at 500 °C and at 700 °C was 148.7 m^2^/g, which was a decrease of 13.5%. The specific surface area of the pure SiO_2_ aerogel decreased by 43.6% in the same conditions. The results indicated that the thermal stability of the Cu-doped aerogel was significantly improved.

Kong et al. [83] synthesized resorcin–formaldehyde (RF)/SiO_2_ composite aerogels using the sol–gel method, with the CO_2_ supercritical fluid drying technique. The C–SiO_2_ aerogels were obtained from the RF/SiO_2_ aerogels after carbonization. The results showed that heat treatment had the least effect on the RF/SiO_2_ aerogel when the RF concentration was 13%. After heat treatment at 1200 °C in a N_2_ atmosphere for 3 h, the specific surface area was still 493 m^2^/g, which means that the aerogel has good thermal stability. However, carbon-containing aerogels will be oxidized in an oxidizing atmosphere.

Zhang et al. [84] also used the sol–gel method with CO_2_ supercritical fluid drying to prepare doped silica aerogels. YCl_3_·6H_2_O and TEOS were used as precursors and the doping concentration of Y_2_O_3_ in the final product ranged from 5 to 30 wt%. The characterization results showed that the Y_2_O_3_–SiO_2_ aerogel maintained the original spatial network structure of the silica aerogel. The thermal stability of the aerogel was improved by adding the yttrium element during the preparation process of the aerogel. Moreover, 10 wt% Y_2_O_3_–SiO_2_ aerogels were maintained in an amorphous state, with a specific surface area of 643.8 m^2^ g^−1^ and an average pore size of 21.3 nm after heat treatment at 900 °C for 2 h.

In summary, heteroatomic (Al, Zr, Y, Cu, and so on) doped silica aerogels showed greater thermal stability than pure silica aerogels. In addition, the properties can be adjusted by altering the ratio of the doping element in regard to silica aerogels in the material composition. The increase in thermal stability results from the formation of more thermally stable Si-O-M (M = Al, Zr, and so on) bonds. On the other hand, the doped phase exhibits superior thermal stability than SiO_2_. In addition, the mechanism of Cu doping, enhancing the thermal stability of silica aerogel samples, needs further study. However, the introduction of other phases can also result in an increase in the thermal conductivity and density of the material, which reduces the thermal insulation performance of the material. Moreover, some elements, such as Zr, in doped silica aerogels, exhibit changes in the nanostructure compared to pure silica aerogels, including significant decreases in the pore size and specific surface area.

### 3.2. Construction of the Surface Heterostructure

The surface energy and hydrophobicity of the surface of silica aerogels can be modified through appropriate surface chemical modification, thereby leading to the improved stability of aerogels [85,86,87]. However, organic surface-modification agents, such as various alkyl-alkoxy/chlorosilane (organosilane) compounds, decompose at high temperatures (exceeding 500 °C) and, thus, do not contribute to the enhanced high-temperature thermal stability of silica aerogels. Inorganic heterostructures, with low surface energy or high thermal stability, constructed on the surface of the silica aerogel backbone matrix, can effectively improve the overall thermal stability of the material. In addition, coating SiO_2_ on the high-temperature resistant aerogel skeleton can also improve the stability of the aerogel.

#### 3.2.1. The Sol–Gel Method

The thermal decomposition of organic surface modifiers typically occurs below 500 °C. Zhang et al. [88] prepared a phenyl-reinforced flexible silica aerogel with high thermal stability using the sol–gel method and the ambient pressure drying technique. The aerogel exhibited a low density of 0.082 g cm^−3^, a high specific surface area of 162.1 m^2^ g^−1^, and a high porosity of 94.2%. The maximum degradation rate temperature was significantly increased, by more than 150 °C, reaching up to 627 °C in air and 742 °C in N_2_.

Inorganic heterostructures can provide higher thermal stability. Si et al. [89] proposed a scalable approach for the synthesis of ceramic nanofibrous aerogels coated with amorphous AlBSi glass ceramics, with a superelastic lamellar structure (Figure 7a). The prepared CNFAs showed excellent overall performance, with a low thermal conductivity of 0.025 W m^−1^ K^−1^, a low density of 0.15 mg cm^−3^, a zero Poisson’s ratio, and rapid recovery at an 80% strain. At 1100 °C, CNFAs can still maintain their elastic resilience well. After heat treatment at 1400 °C, the original skeleton structure of CNFAs can still be maintained.

Tai et al. [92,93] reported on the preparation of titania-coated silica aerogels, incorporating Au nanoparticles as catalysts. The high catalytic activity of the catalysts was preserved after heat treatment at 700 °C for 2 h and the aerogel maintained a good porous nanostructure after heat treatment at 800 °C for 2 h.

In addition to coating other materials on the surface of silica aerogels, the overall thermal stability of the material can also be improved by coating the other aerogel skeleton with SiO_2_. In 2014, Zu et al. [90] developed a new alkoxide chemical liquid deposition technology to prepare MO*_x_*/(MO*_x_*-SiO_2_)/SiO_2_ (M = Al, Zr, Ti) core–shell nanostructured metal oxide aerogels with enhanced strength and thermal stability (Figure 7b). Core–shell nanostructured silica nanoparticles increase stability through MO cores. In addition, the crystal growth and phase transition of the core metal oxides are inhibited by the silica shell. The thermal stability of the core–shell nanostructured aerogels increased from about 400–800 °C to 1000–1300 °C.

Su et al. [91] prepared a SiC@SiO_2_ nanowire aerogel, using a directional thermal conduction freeze casting process (Figure 7c). This aerogel contained nanowires assembled into an array of hierarchical anisotropic structures. The SiC@SiO_2_ nanowire aerogel had excellent properties, including a radial thermal conductivity of 0.014 W m^−1^ K^−1^, full recovery at an 80% strain, and an excellent axial stiffness of 24.7 kN·m kg^−1^. The results of the thermal stability experiments showed that the aerogel can maintain stability at 1200 °C in an air atmosphere. It was found that the excellent thermal protection performance is due to the multi-scale thermal barrier provided by the hierarchical structure. The excellent thermal stability benefits from the high-temperature resistance of the carbide ceramics phase.

The coating of aerogel backbone matrixes with heterogeneous structures was conducted in the gel state during the synthesis process of all the aforementioned works. In fact, the surface coating of the ceramic skeleton structure of silica aerogels is difficult to achieve using liquid phase methods. Moreover, the dimensions of the nanoparticles or aerogel skeleton, synthesized using the liquid method, are relatively large. For example, the diameter of the skeleton in ceramic nanofibrous aerogels (CNFAs) exceeded 200 nm (Figure 7a) and was much larger than common silica aerogels [89], which is not conducive to enhancing the thermal insulation performance of aerogels. In addition, the process involving this method is complicated.

#### 3.2.2. The Thin-Film Deposition Technique

The liquid phase chemical deposition on silica aerogels during the sol–gel stage cannot achieve an accurate adjustment of the interface and cannot target the accurate modification of the skeleton surface. In particular, the skeleton of silica aerogels experiences coarsening and the pore size and specific surface area are reduced. This further leads to an increase in density and thermal conductivity and a decrease in the thermal insulation performance of aerogels. In addition, there are other problems, such as the complicated process, as well as the poor conformal and uniformity of the coating. The precise deposition of a thin film on the nanoparticle skeleton surface is a more feasible solution to improve the thermal stability of aerogels.

The commonly used thin-film deposition techniques include physical and chemical vapor deposition (PVD and CVD) and atomic layer deposition (ALD) [94,95]. However, the surface depositions inside aerogels are complicated due to its high-aspect-ratio internal pore structure [96,97,98,99,100,101]. The ALD technique has emerged as the optimal method for the deposition of high-aspect-ratio materials, due to its inherent repeatability, simplicity, high uniformity, precise control in terms of the composition and thickness, as well as its exceptional conformality at the atomic level (Figure 8a). Table 1 shows a comparison between the three different thin-film vapor deposition techniques [94]. The precise control and regulation of the surface chemical reactions on a substrate is the primary feature of the atomic layer deposition (ALD) technique. ALD distinguishes itself through its transient separation and self-limiting nature during precursor reactions and is an advanced approach for the precise deposition of ultra-thin films at an atomic scale.

The ALD process is a bottom-up approach, with multiple steps as shown in Figure 8b. In the initial step, a substrate is exposed to a gas precursor (i.e., reactant 1). The introduction of the precursor initiates chemisorption and an interaction on the surface of the substrate. Ideally, this process continues until all the surface groups have been fully consumed by the precursor. Subsequently, in the second (2nd) purge step, both these by-products and reactant 1 are removed. In the third step, chemisorbed reactant 1 on the substrate surface reacts with the co-reactant (i.e., reactant 2). The subsequent second purge phase eliminates any unreacted co-reactants and by-products, leading to the formation of a new surface layer with active sites. Iterating this procedure enables precise deposition on the substrate surface to achieve the desired thickness.

Aerogels have nanoscale pores and an extremely high aspect ratio (>10,000), which poses considerable challenges for ALD deposition technology. The achievement of complete internal coating by ALD becomes increasingly difficulty with a decreasing pore size and increasing block thickness (i.e., aspect ratio), due to the limited accessibility of the precursor molecules to the interior surfaces. Many pieces of research have been conducted on ALD-deposition aerogels with the aims of improving the thermal stability of the materials.

Elam et al. [103] successfully deposited Al_2_O_3_, using the ALD technique, in silica gel particles, with a radius of 50 um and a pore size of 30 nm, which exhibited an aspect ratio of approximately 1600. The EDAX measurements performed on the silica gel specimens demonstrated that the silica gel particles were infiltrated with the aluminum from the outside surface to the core progressively (Figure 9a). Kucheyev et al. [104,105] presented a study on the atomic layer deposition of a ZnO layer, with a thickness of 2 nm, onto the inner surface of a nanoporous silica aerogel backbone matrix, which exhibited an exceptionally high aspect ratio of ~10^5^. The aerogels consisted of randomly oriented and interconnected amorphous-SiO_2_ and core-ZnO shell nanoparticles, forming an open network, with a specific surface area of ~100 m^2^g^−1^. The uniformity and crystallinity of the ZnO coating were revealed using secondary ion mass spectrometry (SIMS) and high-resolution TEM. Kucheyev et al. [106] also studied the atomic layer deposition of Cu and Cu_3_N on the inner surfaces of nanoporous silica aerogels, with a ~9 nm pore size (Figure 9b). The maximum penetration depth was 25 μm and 80 μm, respectively, corresponding to an aspect ratio of 2.8 × 10^3^ and 8.9 × 10^3^.

Mane et al. [107] reported on a high-temperature resistant material prepared through the deposition of a tungsten (W) thin film using the ALD technique, using high-surface-area silica aerogels as templates (Figure 9c). The material was prepared using the porogen method and the final product had a density ranging from 0.3 to 0.5 g cm^−3^, the size of the nanopore was 40 nm, and the thickness was about 2 mm. Moreover, Si_2_H_6_ and WF_6_ were sequentially exposed as gas precursors on the inner surface of the silica aerogels at 200 °C. Densities as high as 5 g cm^−3^ were achieved by adjusting the number of W ALD cycles. In addition, the BET measurements showed that the specific surface area of the silica aerogels decreased from 381 m^2^/g to 161 m^2^/g and the average mesopore diameter was reduced from 38.8 nm to 21.3 nm after fifteen W ALD cycles. The nanoporous aerogel coated with the W coating withstood the thermal test of ultra-high temperature heating at 1500 °C in a vacuum, which means that it has great potential for application in regard to solid rare isotope catchers.

Gayle et al. [108] proposed a multidose quasi-static mode (QSM) procedure to deposit conformal thin-film coatings on ultra-high-aspect-ratio silica aerogels (Figure 9d). Specifically, conformal coating was successfully achieved on silica aerogels with a pore size of ~20 nm, an overall thickness of ~2.5 mm, and an aspect ratio of more than 60,000:1, by adjusting the ALD exposure time and precursor dose. The tunable control of the ALD infiltration depth was achieved using the QSM procedure. Additionally, an ALD coating process model was developed to accurately describe the coating process. Finally, high-temperature structural stabilization enabled by the application of a conformal ALD Al_2_O_3_ coating was demonstrated using annealing conditions at 800 °C for silica aerogels. Berquist et al. [109] further synthesized a solar-transparent refractory aerogel (3 mm) using the QSM procedure (Figure 9e). The alumina-deposited silica aerogel sample remained stable at 800 °C in air, which was significantly better in terms of performance than the pure silica aerogel. It was suggested that the improved stability was due to the formation of a high-temperature resistant aluminum silicate phase.

The experimental results of Yang et al. [110] also confirmed that Al_2_O_3_ coating using the ALD technique enhances the thermal stability of silica aerogels. Silica aerogels coated with an Al_2_O_3_ coating shrunk at 600 °C, while silica aerogels without an Al_2_O_3_ coating showed significant shrinkage after heating at 400 °C. After 25 Al_2_O_3_ ALD cycles applied to silica aerogels as a transparent thermal insulation material in solar thermal applications, the transmission of 3 mm thick samples decreased from 98.0% to only 1.7%

As mentioned above, ALD technology excels in the deposition of high-aspect-ratio material coatings and alumina thin films effectively improve the thermal stability of silica aerogel. The controllable ALD infiltration depth is important for ALD on silica aerogels or other high-aspect-ratio materials. Due to the self-limiting reaction mechanism, the aerogel skeleton can be accurately conformally modified using ALD, thus optimizing the aerogel surface, while maintaining the unique structure and properties of the aerogel. However, the ALD deposition depth in nanoporous silica aerogels is still limited to the millimeter level, which restricts its application. Additionally, process adjustments are required to avoid non-ideal ALD deposition behaviors, such as pore and reaction site plugging, the decomposition and depletion of precursors.

## 4. Conclusions

Silica aerogels, a unique nanoporous material, with low thermal conductivity and a high specific surface area, are widely used in thermal insulation systems, including in aerospace, energy, and construction-related applications. However, the thermal stability of silica aerogels has been a limiting factor, as their degradation at high temperatures currently restricts their application to below 600 °C in atmospheric conditions.

This review investigates the driving force in the structural evolution of silica aerogels during heat treatment and the process of structural degradation for various temperature ranges. A series of studies on silica aerogel sintering models are reviewed, which deepen our understanding of optimizing the thermal stability of silica aerogels. Heterogeneous element doping and heterogeneous structure construction are the most convincing and effective strategies to improve the thermal stability of silica aerogels. In recent years, silica aerogels with various compositions and microstructures have been carefully designed and developed to enable them to be used in high-temperature applications, such as thermal insulation in aerospace applications, solar power systems, and in relation to heterogeneous catalysis. The deposition of ultra-thin temperature-resistant coatings on the inner surface of aerogels using the ALD technique can improve the thermal stability of silica aerogels, while maintaining the unique advantages of silica aerogels in terms of their low thermal conductivity and low density, which has great development potential. In the future, the key factors for the development of the high-temperature silica aerogel field will be the simultaneous development of composition optimization, novel heterogeneous structures, and the integration of experimental characterization and computer technology.

## Figures and Tables

**Figure 1 gels-11-00357-f001:**
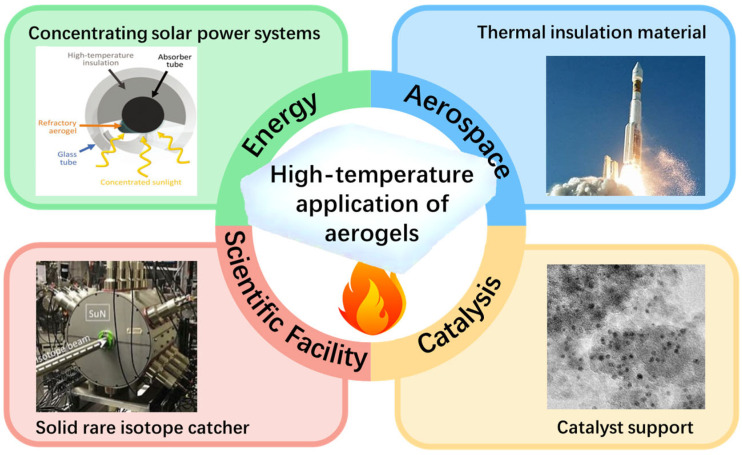
High-temperature applications of aerogels.

**Figure 3 gels-11-00357-f003:**
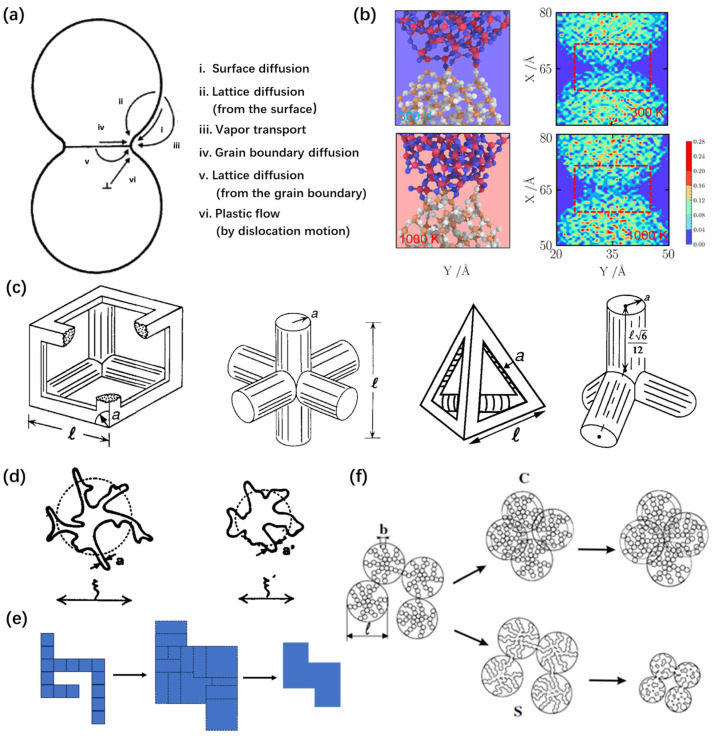
Sintering driving force and model of silica aerogels. (**a**) Schematic of six sintering mechanisms involving adjacent particles. Adapted with permission from [35], copyright 2023, Elsevier. (**b**) Contact region between nanoparticles and number density distributions at 300 K and 1000 K, calculated using MD simulation. Adapted with permission from [28], copyright 2022, Elsevier. (**c**) Schematic and alternative view of cubic and tetrahedra unit cell. Adapted with permission from [36], copyright American Ceramic Society. (**d**) Schematic of an individual aggregate at two different stages of sintering, with a different coherence length *ξ*. Adapted with permission from [37], copyright 1995, Elsevier. (**e**) The dressing procedure showing an original structure (left) with a dressing step (middle) and the recovery step back to the original area (right), using a two-dimensional sketch, consisting of 16 squares [38]. (**f**) Schematic of the solid transport caused by sintering (S) and the motion of the solid network induced by compression (C). Adapted with permission from [39], copyright 2003, Elsevier.

**Figure 4 gels-11-00357-f004:**
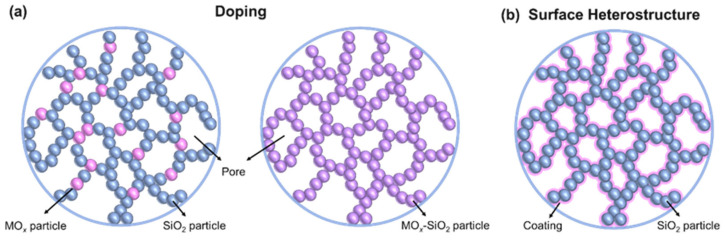
Schematic diagrams of two thermal stability optimization strategies. (**a**) Heteroatom doping. The left side is heterogeneous doping and the right side is homogeneous doping. (**b**) Construction of the surface heterostructure.

**Figure 6 gels-11-00357-f006:**
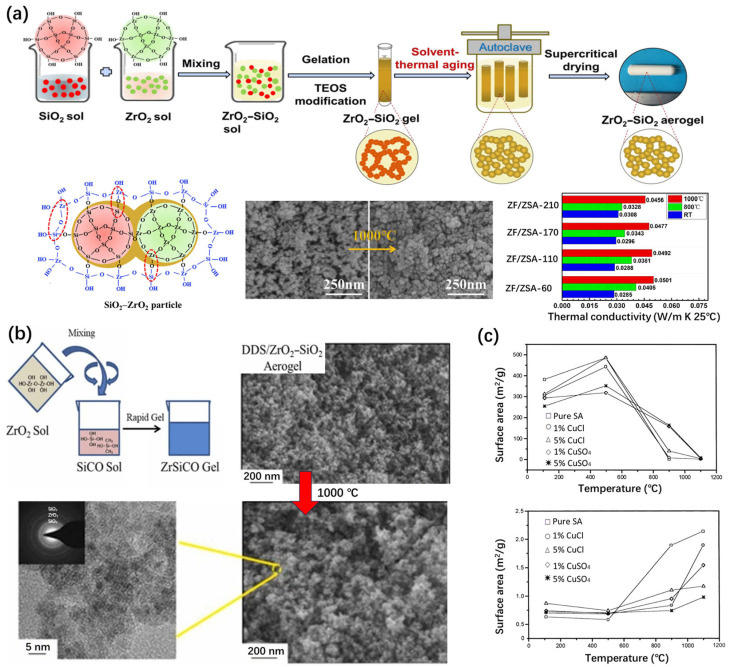
Zirconium or copper-doped silica aerogels with modified thermal stability. (**a**) Synthesis and thermal stability of zirconia-silica aerogels. Adapted with permission from [78], copyright 2020, Elsevier. (**b**) Synthesis and thermal stability of diethoxydimethylsilane (DDS)-modified ZrO_2_–SiO_2_ aerogels. Adapted with permission from [79], copyright 2023, Elsevier. (**c**) Variation in the density and surface area as functions of temperature in pure silica aerogels and Cu-doped silica aerogels. Adapted with permission from [80], copyright 2000, Elsevier.

**Figure 7 gels-11-00357-f007:**
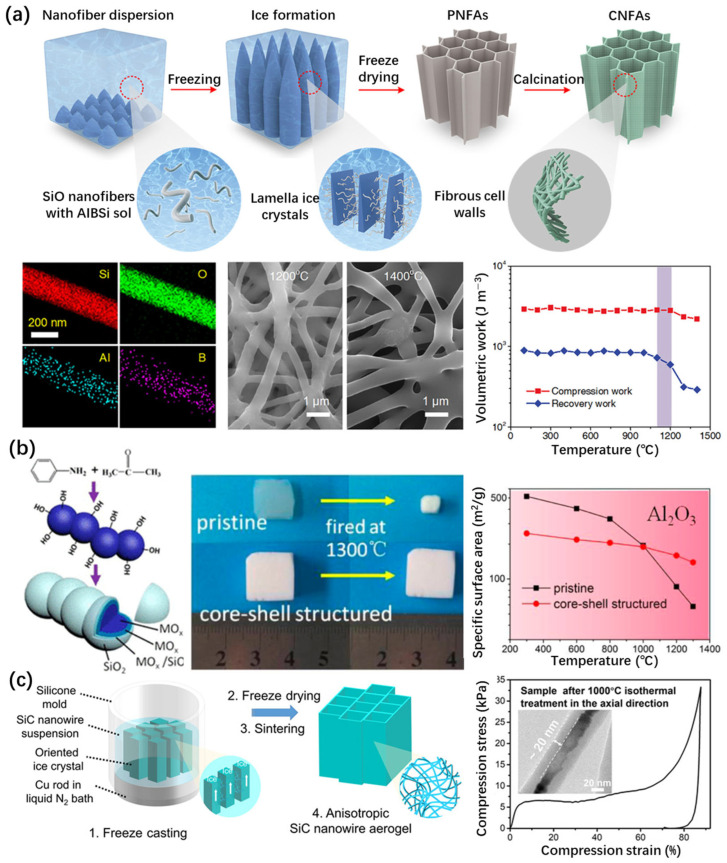
Construction of silica aerogel surface heterostructures using the sol–gel method. (**a**) Synthesis, elemental analysis, and thermal stability of superelastic lamellar-structured amorphous AlBSi glass ceramic–SiO_2_ nanofiber aerogels via the freeze-drying method [89]. (**b**) Synthesis and thermal stability of Al_2_O_3_/(MO_3_-SiO_2_)/SiO_2_ (M = Al, Zr, Ti) core-shell nanostructured aerogels. Adapted with permission from [90], copyright 2014, American Chemical Society. (**c**) Synthesis and thermal stability of a hierarchical anisotropic SiC@SiO_2_ nanowire aerogel fabricated using the controlled directional freeze casting method [91].

**Figure 8 gels-11-00357-f008:**
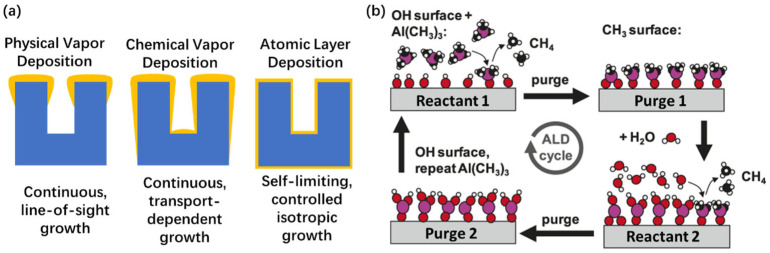
Atomic layer deposition (ALD) technique. (**a**) Schematic diagram of conformality of PVD, CVD, and ALD technique. (**b**) Schematic diagram of an ALD cycle involving aluminum oxide. The use of trimethylaluminum (TMA, Al(CH_3_)_3_) and water sequentially, with the surface groups as reactants, and two reaction steps are separated by inert gas purge steps. Adapted with permission from [102], copyright 2007, American Chemical Society.

**Figure 9 gels-11-00357-f009:**
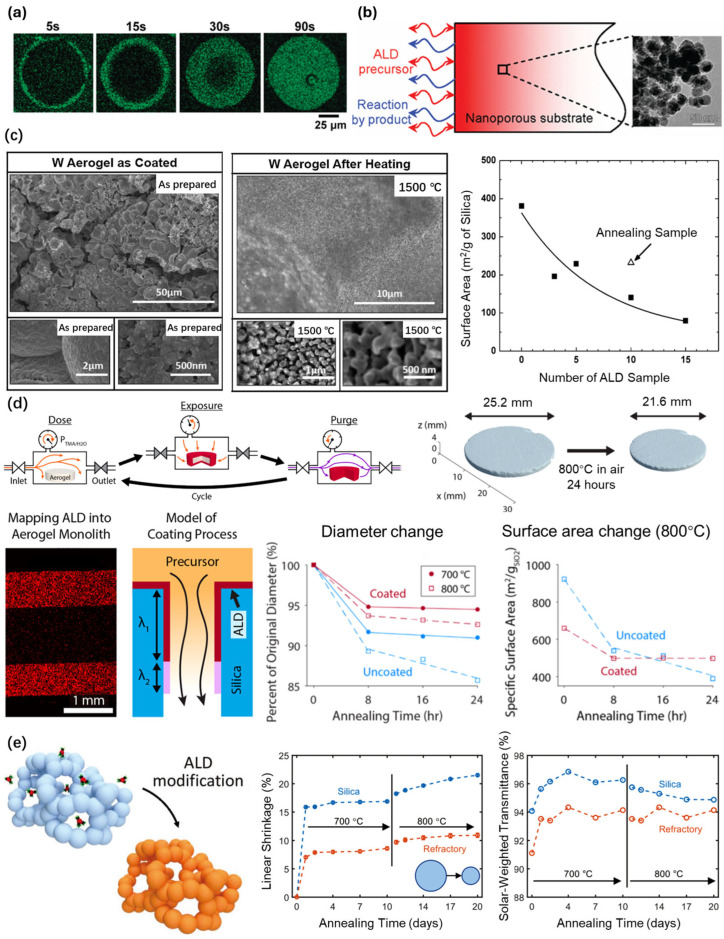
Thin-film deposition on silica aerogels via the ALD technique. (**a**) The distribution of aluminum elements in silica gel particles for different exposure times, obtained from the cross-section of the particles. Adapted with permission from [103], copyright 2010, American Chemical Society. (**b**) The atomic layer deposition of Cu and Cu_3_N on silica aerogels, limited by Knudsen diffusion of the precursor into the pores and the interaction of the precursor with the pore walls. Adapted with permission from [106], copyright 2008, American Chemical Society. (**c**) SEM images of W-deposited silica aerogels before and after 1500 °C annealing in a vacuum, with the change in the surface area. Adapted with permission from [107], copyright 2012, Elsevier. (**d**) Schematic diagram of the two multidose (QSM) steps and the thermal stability of Al_2_O_3_-deposited silica aerogels. Adapted with permission from [108], copyright 2021, American Chemical Society. (**e**) High-temperature resistance of solar-transparent refractory aerogels coated with Al_2_O_3_, using the QSM procedure. Adapted with permission from [109], copyright 2021, Wiley.

**Table 1 gels-11-00357-t001:** Comparison of parameters for three thin-film vapor deposition techniques [94].

Technique	PVD	CVD	ALD
Uniformity	∼80 range	∼10 range	∼80 range
Conformity	<50%	<70%	<100%
Cleanliness	Particle	Particle	No particle
Deposition rate	Fast	Fast	Poor
Vacuum	High	High/Medium	Medium
Temperature range	Low	Low	Wide
Technology	∼100 nm	∼90–65 nm	No limit

## Data Availability

No new data were created or analyzed in this study. Data sharing is not applicable to this article.

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
