# Peer review of "A Review of High-Temperature Resistant Silica Aerogels: Structural Evolution and Thermal Stability Optimization"

_gels, 2025, doi:10.3390/gels11050357_

Round 1
Reviewer 1 Report
Comments and Suggestions for Authors
The manuscript, “A Review of High Temperature Resistant Silica Aerogels: Structure Evolution and Thermal Stability Optimization,” provides a thorough and valuable analysis of silica aerogels, particularly their structural evolution and thermal stability in high-temperature applications. The paper addresses an important topic, as silica aerogels are critical in advanced thermal insulation systems. While the manuscript is well-researched and organized, there are areas requiring minor revisions to improve its clarity, precision, and overall quality.
One notable issue involves content-related ambiguities. For instance, the statement “Silica aerogels are extensively used for super thermal insulation, adsorption catalysis, and energy application” is overly general and lacks specific examples or supporting references. It would be more informative to mention concrete applications, such as thermal insulation in aerospace or adsorption processes for environmental applications, supported by relevant sources. Similarly, the section describing the sintering process states, “The sintering process of silica aerogels results in the degradation of their pore structure,” but does not provide sufficient detail on the mechanisms leading to pore collapse or the subsequent impact on thermal conductivity. Expanding this explanation would strengthen the scientific rigor of the discussion. Additionally, the comparison between the melting points of zirconia (2715 °C) and silica (1715 °C) may be misleading, as nanoscale phenomena like surface diffusion and bond stability also contribute to thermal performance. This comparison should be clarified to reflect the broader factors influencing high-temperature resistance.
Language-related errors appear throughout the manuscript, though they are minor and do not significantly obstruct comprehension. For example, incorrect article usage can be seen in sentences like “The silica aerogel has great development potential,” which should be corrected to “Silica aerogels have great development potential.” There are also inconsistencies in verb tenses; for instance, “Many researchers have studied the structural evolution of silica aerogels” could be revised to “Many researchers study the structural evolution of silica aerogels” to reflect ongoing research trends. Improper punctuation is another recurring issue, such as in the sentence “Silica aerogels are lightweight, porous materials, they are used for…” which should be split for clarity: “Silica aerogels are lightweight, porous materials. They are used for...” Finally, redundancy appears in phrases like “The surface energy of the materials can be effectively reduced by constructing heterogeneous interfaces, thereby impeding the mass transfer process of surface diffusion under high temperature conditions,” which could be simplified to “Reducing surface energy through heterogeneous interfaces limits mass transfer during high-temperature diffusion.”
From a stylistic perspective, some sentences are overly technical and unnecessarily complex. For example, “The crosslinking of weakly branched ZrOâ‚‚ and SiOâ‚‚ clusters can make the mixture of ZrOâ‚‚/SiOâ‚‚ the most uniform and show the best thermal stability” could be simplified to “The uniform crosslinking of ZrOâ‚‚ and SiOâ‚‚ clusters enhances the thermal stability of the material.” Additionally, certain sections, such as the one on sintering models, are excessively detailed, making the text dense and challenging to follow. Summarizing the key findings of these models while referring readers to primary literature for further details would improve the flow and accessibility of the discussion. Inconsistent terminology is another issue, with terms like “high temperature resistance,” “thermal stability,” and “sintering resistance” being used interchangeably. Standardizing these terms throughout the text would improve clarity.
To further enhance the manuscript, the abstract should be revised to present the key findings and future directions concisely, avoiding overly technical descriptions. Visual aids such as schematic diagrams comparing heteroatom doping strategies and surface modification methods would improve comprehension and engagement. Simplifying technical explanations and ensuring consistency in sentence structure, verb tense, and terminology would also make the manuscript more readable.
In summary, while the manuscript offers a comprehensive and valuable review of thermal stability improvements in silica aerogels, minor revisions are required to address content-related ambiguities, language inaccuracies, and stylistic issues. Simplifying overly technical sections and improving clarity will enhance the overall quality and accessibility of the paper. With these revisions, the manuscript will meet high publication standards.
Author Response
Comment 1: The manuscript, “A Review of High Temperature Resistant Silica Aerogels: Structure Evolution and Thermal Stability Optimization,” provides a thorough and valuable analysis of silica aerogels, particularly their structural evolution and thermal stability in high-temperature applications. The paper addresses an important topic, as silica aerogels are critical in advanced thermal insulation systems. While the manuscript is well-researched and organized, there are areas requiring minor revisions to improve its clarity, precision, and overall quality.
Response 1: Thank you for the recommendation and constructive comments. The manuscript has been revised according to the comments.
Comment 2: One notable issue involves content-related ambiguities. For instance, the statement “Silica aerogels are extensively used for super thermal insulation, adsorption catalysis, and energy application” is overly general and lacks specific examples or supporting references. It would be more informative to mention concrete applications, such as thermal insulation in aerospace or adsorption processes for environmental applications, supported by relevant sources.
Response 2: Thanks for this wise suggestion. The manuscript has been revised according to the comments.
In Abstract, we replaced “Silica aerogels are extensively used for super thermal insulation, adsorption catalysis, and energy application” with “Silica aerogels are extensively used in thermal insulation of aerospace and building construction, adsorption processes for environmental applications, concentrating solar power systems and so on” in the revised manuscript.
In the introduction, the specific application of silica aerogel was mentioned in more detail, such as “The low thermal conductivity of silica aerogels makes them suitable for insulation purposes in building construction [14,15], concentrating solar power devices [16], and aeronautics, aerospace systems [17]” and “In addition, The superhydrophobic silica aerogels exhibit remarkable efficacy as reusa-ble absorbents for oils and organic liquids, boasting a high uptake capacity and efficiency [9]. The utilization of silica aerogels as photoanodes for dye sensitized solar cells is attributed to their exceptional surface area and porosity [10]. The low dielectric constant of silica aerogels makes them suitable for use as intermetal dielectric shield-ing materials in microelectronic devices [11]”, supported by reference.
Comment 3: Similarly, the section describing the sintering process states, “The sintering process of silica aerogels results in the degradation of their pore structure,” but does not provide sufficient detail on the mechanisms leading to pore collapse or the subsequent impact on thermal conductivity. Expanding this explanation would strengthen the scientific rigor of the discussion.
Response 3: Thanks for your constructive suggestion. We replaced “The sintering process of silica aerogels results in the degradation of their pore structure and a consequent reduction in specific surface area, thereby compromising the thermal insulation performance of silica aerogels” with “The sintering process of silica aerogels, driven by the reduction of surface energy, intensifies particle aggregation and skeleton coarsening, leading to densification and degradation of their nanopore structure. This structural degradation results in increased solid thermal conductivity, thereby compromising the thermal insulation performance of silica aerogels” in the revised manuscript.
A more detailed introduction of the mechanism, specific process and driving force of pore collapse and structural degradation was discussed in section 2.1 and 2.2.
Comment 4: Additionally, the comparison between the melting points of zirconia (2715 °C) and silica (1715 °C) may be misleading, as nanoscale phenomena like surface diffusion and bond stability also contribute to thermal performance. This comparison should be clarified to reflect the broader factors influencing high-temperature resistance.
Response 4: Thank you for this valuable comment. We have removed the comparison about the melting point of zirconia and silica, which is not the only factor in high temperature resistance.
Replace “The melting point of zirconia is as high as 2715 °C, which is higher than that of silicon oxide at 1715 °C. It can be seen that zirconia has a higher temperature resistance than silicon oxide” with “Zirconia aerogels are potential candidates for high-temperature applications, attributed to the high melting point of ZrO2 at 2715 °C” in section 3.1.2.
Comment 5: Language-related errors appear throughout the manuscript, though they are minor and do not significantly obstruct comprehension. For example, incorrect article usage can be seen in sentences like “The silica aerogel has great development potential,” which should be corrected to “Silica aerogels have great development potential.”
Response 5: We sincerely thank you for careful reading. We have corrected the “The silica aerogel …” into “Silica aerogels …” in Abstract, Introduction, section 2.1, section 3.2, section 3.2.2 and Conclusions.
Comment 6: There are also inconsistencies in verb tenses; for instance, “Many researchers have studied the structural evolution of silica aerogels” could be revised to “Many researchers study the structural evolution of silica aerogels” to reflect ongoing research trends. Improper punctuation is another recurring issue, such as in the sentence “Silica aerogels are lightweight, porous materials, they are used for…” which should be split for clarity: “Silica aerogels are lightweight, porous materials. They are used for...” Finally, redundancy appears in phrases like “The surface energy of the materials can be effectively reduced by constructing heterogeneous interfaces, thereby impeding the mass transfer process of surface diffusion under high temperature conditions,” which could be simplified to “Reducing surface energy through heterogeneous interfaces limits mass transfer during high-temperature diffusion.”
Response 6: We sincerely thank you for careful reading. We corrected the improper tense, punctuation and redundancy based on your kind suggestion.
Comment 7: From a stylistic perspective, some sentences are overly technical and unnecessarily complex. For example, “The crosslinking of weakly branched ZrOâ‚‚ and SiOâ‚‚ clusters can make the mixture of ZrOâ‚‚/SiOâ‚‚ the most uniform and show the best thermal stability” could be simplified to “The uniform crosslinking of ZrOâ‚‚ and SiOâ‚‚ clusters enhances the thermal stability of the material.”
Response 7: Thank you for the comment. Sentences have been simplified according to your suggestion.
Comment 8: Additionally, certain sections, such as the one on sintering models, are excessively detailed, making the text dense and challenging to follow. Summarizing the key findings of these models while referring readers to primary literature for further details would improve the flow and accessibility of the discussion.
Response 8: Thank you for the comment. A great deal of detail has been omitted in section 2.2, section 3.1.2 and section 3.2.1. Only the main findings of these models and studies are retained.
Comment 9: Inconsistent terminology is another issue, with terms like “high temperature resistance,” “thermal stability,” and “sintering resistance” being used interchangeably. Standardizing these terms throughout the text would improve clarity.
Response 9: Thank you for your constructive suggestions. We unify the expression by replacing all “high temperature resistance” and “sintering resistance” with “thermal stability”, except when representing properties specific to applications.
Comment 10: To further enhance the manuscript, the abstract should be revised to present the key findings and future directions concisely, avoiding overly technical descriptions.
Response 10: Thanks for the kindly comment. The abstract was simplified and revised by your suggestions.
Revised version: “Silica aerogels are exhibit exceptionally low thermal conductivities and low apparent densities as the unique porous nanomaterial. They are extensively used in thermal insulation of aerospace and building construction, adsorption processes for environmental applications, concentrating solar power systems and so on. However, the degradation of silica aerogel nanoporous structure at high temperatures seriously restrict their practical applications. Through a comprehensive review of the high-temperature structural evolution and sintering mechanisms of silica aerogels, this paper introduces two strategies to enhance their thermal stability, including heteroatom doping and surface heterogeneous structure construction. In particular, atomic layer deposition (ALD) of ultra-thin coatings on silica aerogel holds significant potential for enhancing thermal stability while preserving its ultra-low thermal conductivity.”
Original version: “Silica aerogels are exhibit exceptionally low thermal conductivities and low apparent densities as the unique porous nanomaterial. Silica aerogels are extensively used for super thermal insulation, ad-sorption catalysis and energy application. However, the silica aerogels have a defect in terms of the degradation of porous structure and thermal insulation at high temperatures, which seriously re-strict their practical applications. The scientific achievements of the structure evolution and sin-tering mechanism of silica aerogels are showed. Improving the temperature resistance of aerogel particles and increasing the thermal stability of the aerogel backbone surfaces can effectively inhibit the sintering of materials. This review encompasses a complete survey of temperature resistance optimization of silica aerogels through heteroatom doping and surface heterogeneous structure construction. This review aims to provide a thorough introduction and deep understanding of the mechanism and methods for thermal stability optimization of silica aerogels. In the future, Silica aerogel has great development potential in high temperature applications with a greater thermal stability while maintaining the ultra-low thermal conductivity.”
Comment 11: Visual aids such as schematic diagrams comparing heteroatom doping strategies and surface modification methods would improve comprehension and engagement. Simplifying technical explanations and ensuring consistency in sentence structure, verb tense, and terminology would also make the manuscript more readable.
Response 11: Thank you for the comment. Schematic diagrams of two thermal stability optimization strategies were added as Figure 4.
Figure 4. Schematic diagrams of two thermal stability optimization strategies. (a) Heteroatom doping. The left side is heterogeneous doping, and the right side is homogeneous doping. (b) Construction of surface heterostructure.
Comment 12: In summary, while the manuscript offers a comprehensive and valuable review of thermal stability improvements in silica aerogels, minor revisions are required to address content-related ambiguities, language inaccuracies, and stylistic issues. Simplifying overly technical sections and improving clarity will enhance the overall quality and accessibility of the paper. With these revisions, the manuscript will meet high publication standards.
Response 12: We appreciate your time and effort for your insightful and constructive comments. We are convinced that your comments have greatly improved our manuscript and brought us a lot of inspiration. If there are any other modifications we could make, we would like very much to modify them and we really appreciate your help.
Reviewer 2 Report
Comments and Suggestions for Authors
Reading the manuscript and also another, in the same journal earlier in 2024 published paper entitled: “A review of High temperature Aerogels: Composition, Mechanism and Properties”, I was surprised that the authors do not seem to know this paper and the content. I am questioning, why the same topic merits two publications in the same journal in such a short timely distance. Obviously the authors are not aware of the other manuscript. I need to admit, that the questions/subject are both manuscripts are not entirely the same, however , there is a good part of overlap between them. At least the part of the mentioned article dealing with silica aerogels and with silica based composites as well as the part dealing with silica nanofiber composite aerogels have also a part in the submitted manuscript. AS mentioned, a critical discussion of the things already published in the previous article is not present, also the references given in this article are only to a minor part referenced and discussed in the submitted manuscript. This is especially important because the authors try to encompass a complete survey of temperature resistance optimization of silica aerogels. Rightly heteroatom doping and surface heterogeneous structures are discussed, however he claimed deep understanding of methods for thermal stability optimization is only partially achieved.
The manuscript has certainly a value while discussing the structural evolution during high temperature treatment as well as the sintering mechanism and consequently (with the discussion of doping) also mechanisms involve to stabilize structures and to prevent at least partially collapse while heating aerogels. Surprising is then the discussion of e.g. Cu doping which obviously does not really work. The stabilization with nanofibers is really only briefly discussed, also possibilities to generate flexible structures by e.g. silica nanofibers. The discussion of thin film techniques is interesting, however with the small pore size limiting the mean free path of air not the best method for application to improve properties of silica aerogels.
In summary, the authors are invited to re-write their manuscript in view of the already existing article(s) and to concentrate on subject areas not tackled in other, recently published works.
Author Response
Comment 1: Reading the manuscript and also another, in the same journal earlier in 2024 published paper entitled: “A review of High temperature Aerogels: Composition, Mechanism and Properties”, I was surprised that the authors do not seem to know this paper and the content. I am questioning, why the same topic merits two publications in the same journal in such a short timely distance. Obviously the authors are not aware of the other manuscript. I need to admit, that the questions/subject are both manuscripts are not entirely the same, however , there is a good part of overlap between them. At least the part of the mentioned article dealing with silica aerogels and with silica based composites as well as the part dealing with silica nanofiber composite aerogels have also a part in the submitted manuscript. AS mentioned, a critical discussion of the things already published in the previous article is not present, also the references given in this article are only to a minor part referenced and discussed in the submitted manuscript. This is especially important because the authors try to encompass a complete survey of temperature resistance optimization of silica aerogels. Rightly heteroatom doping and surface heterogeneous structures are discussed, however he claimed deep understanding of methods for thermal stability optimization is only partially achieved.
Response 1: Thank you for the constructive comments. In fact, we were already aware of the article (Wang et al. Gels 2024, 10(5), 286) published in Gels when we submitted our manuscripts. The article made a comprehensive overview of the five classes of aerogels, detailing their respective doping elements, thermal insulation mechanisms, and a wide range of practical applications. As for silica-based aerogel, Wang et al. introduced the improvement of the performance of silicon-based aerogel, including strength, toughness, and high-temperature resistance. However, the improvement method introduced in this paper was mainly the composite of aerogel materials. For example, silica aerogels were composite with spherical, hollow infrared shading agent, glass fiber felt, mullite fiber felt, ceramic nanofiber and so on.
Therefore, the main content of our manuscript is not the discussion of composite aerogel, and focuses mainly on the improvement of aerogel composition and structure itself. Indeed, silica-based aerogel composites and silica nanofiber composite aerogels are effective strategies to improve the high temperature resistance of thermal insulation materials in practical applications. However, our manuscript is based on the mechanism of silica-based aerogel structural evolution. The improvement of temperature resistance of silica-based aerogel itself rather than that of composites is discussed in detail. This explains why we did not cite this paper in our initial manuscript submission. We sincerely believe that these two articles can be published in the same journal in a short time. It also reflects the professionalism of this special issue.
In the original manuscript we mentioned, “Although the development of composite aerogels can indeed improve the high temperature resistance by introducing of refractory, infrared shading agent and structural design, composite aerogels are not within the scope of silica aerogels in this review. This review introduces the structure evolution and thermal insulation failure mechanism of silica aerogels at high temperature”. For example, if the subject of the previous article was "A Review of Global Culture", the subject of our manuscript is "A Review of Asian Food Culture".
The surface heterostructure constructed by sol-gel method in section 3.2.1 was also essentially a composite aerogel, which was the part where our manuscripts overlap with Wang’s article. However, surface modifications are different from the composite of aerogel with additive phases such as quartz fiber felt, mullite felt and so on, which still retains the macroscopic and microscopic structure of aerogel. The reference did not distinguish between the two types of composites. In addition, the section of surface heterostructure with sol-gel method was meant to lead out the ultra-thin coating deposition on silica aerogel surface structure in the next section, which composites was not the focus of this manuscript.
Based on your insightful suggestions, we recognize that there is a possibility for readers to misunderstand the overlapping content between two articles. Therefore, we added the following part in the introduction to explain the differences between the two articles, and quoted Wang’s article: "Various strategies have been explored to overcome this limitation, including chemical doping, surface modification and composite materials. Wang et al. (Wang et al. Gels 2024, 10(5), 286) comprehensively reviewed the five types of high-temperature aerogels including polyimide-based, zirconia-based, silica-based, alumina-based and carbon-based aerogels. In the section on silica-based aerogels, the authors provide a concise review of performance optimization referring to thermal conductivity, strength, toughness, and high-temperature resistance. The focus is primarily on composite aerogels. This review mainly discusses the improvement of high temperature resistance of aerogel structure itself, including doping and surface modification. In particular, the structure evolution and thermal insulation failure mechanism of silica aerogel at high temperature are introduced. Based on the mechanism, the methods developed to improve the thermal stability of silica aerogels were systematically examined, with a focus on their underlying mechanisms, recent advancements, and challenges."
Comment 2: The manuscript has certainly a value while discussing the structural evolution during high temperature treatment as well as the sintering mechanism and consequently (with the discussion of doping) also mechanisms involve to stabilize structures and to prevent at least partially collapse while heating aerogels. Surprising is then the discussion of e.g. Cu doping which obviously does not really work. The stabilization with nanofibers is really only briefly discussed, also possibilities to generate flexible structures by e.g. silica nanofibers. The discussion of thin film techniques is interesting, however with the small pore size limiting the mean free path of air not the best method for application to improve properties of silica aerogels.
Response 2: Thanks for your constructive comments. In fact, the mechanism of Cu doping enhancing the thermal stability of silica aerogel samples needs further study. The clarification of this research gap is supplemented in the last paragraph of section 3.1.3.
As for the stability of nanofibers to aerogel composites, our paper does not focus on the discussion, so as to avoid overlapping with previous articles. It serves only as an introduction to the discussion of thin film deposition technique in the next section.
Comment 3: In summary, the authors are invited to re-write their manuscript in view of the already existing article(s) and to concentrate on subject areas not tackled in other, recently published works.
Response 3: Thank you again for your time and meaningful suggestions. Based on your suggestions, we believe our revised manuscript has been greatly improved. If there are any other modifications we could make, we would like very much to modify them and we really appreciate your help.
Reviewer 3 Report
Comments and Suggestions for Authors
The paper is well structured and well written. The idea to discuss the mechanism of sintering of aerogel structures and to introduce the reader into the mechanism of thinking how to stabilize the structure and make the material start morphological changes at higher temperatures is well presented. The authors display the possibility to use other oxides to modify the structure and stabilize it and to use the doping of silica to make the material more temperature resistant. Finally the way to prepare structures that are effectively two layered by introducing another layer over the scaffold made of silica is well presented and interesting to read.
I would only be concerned about the way how authors address the chemical composition of studied gels. Sometimes they address the compositional modification like the addition of aluminum and sometimes of alumina. My opinion is that it is alumina as in the working environment the material will turn to oxide. The same is for other metal doping elements, as the Si-O-M bonds are created I feel that one should talk about the oxides that modify the structure of another oxide, here silica, and not individual metals.
Some minor changes are following and are really mostly typo or the way how to address the composition as added metals or their oxides.
In the phrase authors should add °C after 600
The results revealed that at both 400 °C and 600, the temperature-dependent changes reached a plateau within a month.
Page 4 the paragraph about the fractal dimension should have a reference where the data came from. I guess that it is the same as the following one, but just put it in the text. And some images illustrating the fractal analysis approach would be interesting to see, but not compulsory for this paper.
On page 10 authors write
aluminum-doped silicon aerogels
I would use alumina and silica instead of aluminum and silicon for aerogels as this is better for understanding the real composition and what happens in the material.
Authors didn't mention the thermal conductivity of alumina doped silica gel if some data are available it would be interesting to present them.
Paragraph 3.1.2.
Zirconium doped gel would be more appropriate zirconia doped gel.
Page 19 I suggest to say silica aerogel instead of silicon aerogel.
As mentioned above, ALD technology excels in the deposition of high aspect ratio material coatings and alumina thin film effectively improves the temperature resistance of silicon aerogel.
Author Response
Comment 1: The paper is well structured and well written. The idea to discuss the mechanism of sintering of aerogel structures and to introduce the reader into the mechanism of thinking how to stabilize the structure and make the material start morphological changes at higher temperatures is well presented. The authors display the possibility to use other oxides to modify the structure and stabilize it and to use the doping of silica to make the material more temperature resistant. Finally the way to prepare structures that are effectively two layered by introducing another layer over the scaffold made of silica is well presented and interesting to read.
Response 1: Thank you for the insightful comments on this manuscript. The manuscript has been revised according to your suggestions.
Comment 2: I would only be concerned about the way how authors address the chemical composition of studied gels. Sometimes they address the compositional modification like the addition of aluminum and sometimes of alumina. My opinion is that it is alumina as in the working environment the material will turn to oxide. The same is for other metal doping elements, as the Si-O-M bonds are created I feel that one should talk about the oxides that modify the structure of another oxide, here silica, and not individual metals.
Response 2: Thank you for this valuable comment. In the description of the chemical composition of the aerogel, we have replaced “aluminum” with “alumina” and replaced “zirconium” with “zirconia”.
Comment 3: Some minor changes are following and are really mostly typo or the way how to address the composition as added metals or their oxides. In the phrase authors should add °C after 600. The results revealed that at both 400 °C and 600, the temperature-dependent changes reached a plateau within a month.
Response 3: Thank you for your careful reading. We have made modifications according to this suggestion.
Comment 4: Page 4 the paragraph about the fractal dimension should have a reference where the data came from. I guess that it is the same as the following one, but just put it in the text. And some images illustrating the fractal analysis approach would be interesting to see, but not compulsory for this paper.
Response 4: Thank you for the kindly comment. Your conjecture is correct. We have added the reference into the text.
Comment 5: On page 10 authors write "aluminum-doped silicon aerogels". I would use alumina and silica instead of aluminum and silicon for aerogels as this is better for understanding the real composition and what happens in the material.
Response 5: Thank you for the comment. “Aluminum” and “silicon” have been replaced with “alumina” and “silica”.
Comment 6: Authors didn't mention the thermal conductivity of alumina doped silica gel if some data are available it would be interesting to present them.
Response 6: Thanks for your constructive suggestion. The thermal conductivity of alumina doped silica aerogels has been inserted in the first paragraph of section 3.1.1.
The aluminum-doped silica aerogels maintain low thermal conductivity. In the study by Ling et al., silica-based aerogels doped with 1.28 to 7.46 wt% alumina exhibit thermal conductivities of 0.030 to 0.039 W m-1 K-1 at room temperature, which increase to 0.057 to 0.074 W m-1 K-1 at 800 °C (Ling et al. J Sol-Gel Sci Technol 2018, 87, 83–94).
Comment 7: Paragraph 3.1.2. Zirconium doped gel would be more appropriate zirconia doped gel.
Page 19 I suggest to say silica aerogel instead of silicon aerogel.
Response 7: Thank you for the comment. “Aluminum” and “silicon” have been replaced with “alumina” and “silica”.
Comment 8: As mentioned above, ALD technology excels in the deposition of high aspect ratio material coatings and alumina thin film effectively improves the temperature resistance of silicon aerogel.
Response 8: We appreciate your time and effort for your insightful and constructive comments. We are convinced that your comments have greatly improved our manuscript and brought us a lot of inspiration. If there are any other modifications we could make, we would like very much to modify them and we really appreciate your help.
Round 2
Reviewer 2 Report
Comments and Suggestions for Authors
The authors reacted adequately to my comments and adjusted the manuscript accordingly. Acceptation is recommended. However, there is another small remark and room for improvement, many of the figures are in fact combined figures. In order to increase readability (with the combination of figures in a number of the original plates is the letter size that much reduced, that reading is not any more possible) the authors are requested to have a critical look into this issue and to adjust their figures accordingly.
Author Response
Comments 1: The authors reacted adequately to my comments and adjusted the manuscript accordingly. Acceptation is recommended. However, there is another small remark and room for improvement, many of the figures are in fact combined figures. In order to increase readability (with the combination of figures in a number of the original plates is the letter size that much reduced, that reading is not any more possible) the authors are requested to have a critical look into this issue and to adjust their figures accordingly.
Response 1: Thank you for your constructive suggestions. According to your suggestion, we have modified the picture and enlarged the font in the figure 2a, 2c,2e, figure 3a, figure 5c, 5d, 5e, figure 6a, 6b, 6c, figure 7a, 7b, 7c and figure 9c, 9d, 9e. In addition, Figure 9 has been reformatted for readability.